# Electric Potential of *Chlorella* sp. Microalgae Biomass in Microbial Fuel Cells (MFCs)

**DOI:** 10.3390/bioengineering12060635

**Published:** 2025-06-11

**Authors:** Rickelmi Agüero-Quiñones, Magaly De La Cruz-Noriega, Walter Rojas-Villacorta

**Affiliations:** 1Escuela de Ingeniería Ambiental, Facultad de Ingeniería, Universidad César Vallejo, Trujillo 13007, Peru; 2Institutos y Centros de Investigación, Universidad César Vallejo, Trujillo 13001, Peru; mdelacruzn@ucv.edu.pe; 3Programa de Investigación Formativa e Integridad Científica, Universidad César Vallejo, Trujillo 13001, Peru; wrojasv@ucv.edu.pe

**Keywords:** *Chlorella* sp., microbial fuel cells, biomass, electric power

## Abstract

The projected global energy demand for 2050 drives the imperative search for alternative and environmentally friendly energy sources. An emerging and promising alternative is microbial fuel cells assisted with microalgae. This research evaluated the potential of *Chlorella* sp. biomass in electricity production using microbial fuel cells (MFCs) with a single chamber and activated carbon and zinc electrodes at the laboratory scale over 20 days of operation. Maximum values of voltage (1271 ± 2.52 mV), current (4.77 ± 0.02 mA), power density (247.514 mW/cm^2^), current density (0.551 mA/cm^2^), and internal resistance (200.83 ± 0.327 Ω) were obtained. The biomass-maintained pH values of 7.32 ± 0.03–7.74 ± 0.02 and peaks of electrical conductivity of 2450 ± 17.1 µS/cm and oxidation-reduction potential of 952 ± 20 mV were reached. Meanwhile, cell density and absorbance increased to average values of 2.2933 × 10^7^ ± 1.15 × 10^6^ cells/mL and 3.471 ± 0.195 absorbance units (AU), respectively. Scanning electron microscopy micrographs allowed the observation of filamentous structures of the formed biofilm attached to carbon particles, and energy-dispersive X-ray spectroscopy spectra of the anodes determined the predominance of oxygen, carbon, silicon, aluminum, and iron. Finally, this research demonstrates the great potential of *Chlorella* sp. biomass for sustainable bioelectricity generation in MFCs.

## 1. Introduction

The global energy demand has experienced exponential growth in recent decades, primarily driven by industrial development and population increase [1]. According to projections, the current world population stands at 7.7 billion people and is expected to reach 9.7 billion by 2050 [2], translating into a 50% increase in global energy demand [3]. The majority of this demand is still met through the use of fossil fuels such as coal, oil, and natural gas [4]. However, these resources are finite, and their use has a negative environmental impact due to greenhouse gas emissions, significantly contributing to the phenomenon of global warming [5]. Additionally, the extraction and processing of these fuels can have devastating consequences for ecosystems. 

In this context, the search for alternative and sustainable energy sources becomes imperative to meet our future energy needs in a more environmentally friendly manner [6]. Currently, the global landscape is undergoing a period of energy revolution and transition, where the focus is progressively shifting from fossil fuels to renewable energy [7]. This energy transition involves a significant structural change in how energy will be produced and consumed, aiming to mitigate climate change by increasing the energy contribution from renewable sources such as wind, solar, hydroelectric, geothermal, and biomass [8]. To harness these sources, it is crucial to adopt sustainable energy technologies, among which bioelectrochemical systems (BESs) have attracted significant attention in recent years, with microbial fuel cells (MFCs) being an emerging and promising approach in green electricity generation [9,10]. These cells directly convert chemical energy into electrical energy through electroactive microorganisms [11], which degrade organic compounds and release electrons at the anode, which are transferred through an external circuit to the cathode, producing energy [12]. The concept of MFCs emerged in 1911 when Potter observed the phenomenon of electron transfer by an electromotive force (microorganisms) for electricity production using platinum electrodes immersed in *Escherichia coli* and *Saccharomyces cerevisiae* cultures in a battery-like configuration [13]. This was a significant milestone in the history of MFCs, laying the groundwork for understanding how microorganisms can be used to generate electricity from organic matter oxidation [14]. However, this technology was not extensively explored until the 1990s when its applications gained momentum in wastewater treatment, environmental remediation, and sustainable energy generation [15]. 

The main components of MFCs are the anode, the cathode, the microbial community, and, in some cases, the selective permeable membrane [16]. The anode carries out substrate oxidation (anaerobic), while the cathode undergoes oxidant reduction (aerobic or anaerobic). The microbial community acts as an enzymatic catalyst, and the selective permeable membrane serves as a separator between both chambers, facilitating ion passage [17]. Additionally, their configuration can be single-chambered, double-chambered, or stacked [18]. In terms of economics, efficiency, and reproducibility, single-chamber cells prove to be the most viable option for application; however, the correct configuration often depends on conditions, materials, substrate, and exoelectrogenic microorganisms [19]. 

Alternatively, microalgae-assisted MFCs (A-MFCs) have made significant advances in electricity generation, pollutant removal, wastewater treatment, and biofuel generation [20]. Microalgae biomass contains high levels of proteins (32%) and carbohydrates (51%), which are easily degradable by electrogenic microorganisms to generate electricity [21]. Furthermore, oxygen (O_2_) generated by microalgae through photosynthesis provides a large number of electron acceptors (biocatalysts), continually providing the electromotive force for anode microorganisms to degrade organic matter [22], which results in a synergistic metabolism between bacteria and microalgae to enhance cell electrical production efficiency [23]. These phototrophic organisms require light, carbon dioxide (CO_2_), and nutrients for growth, are found in most aquatic ecosystems, and can live and adapt to various environmental conditions [24]. In addition, they produce considerable biomass in a short period, have a higher capacity for CO_2_ fixation and pollutant removal, can be modified through bioengineering, and possess an energy conversion efficiency rate of 9% [25,26]. All these properties make them a prominent candidate for large-scale use in various applications with MFCs.

Several microalgae species have been identified and used in various research studies to generate electricity in MFCs in recent years, with species of the genus *Chlorella* being one of the most studied [27], although there are also good reports of other species. For instance, Deng et al. (2022) worked with carbon paper electrodes in double-chamber MFCs inoculated with *Chlorella* sp. biomass in the cathode and wastewater in the anode, producing a maximum voltage of 220 mV and a 53.64% removal of Thiacloprid (THI) in 12 days of operation [28]. Similarly, Varanasi et al. (2020) combined the benefits of *Chlorella sorokiniana* microalgae and double-chamber MFCs with carbon felt electrodes to form the carbon capture microbial cell system (MCC), generating a peak voltage of 637 mV, a maximum power density of 2.32 W/m^3^, and a maximum algal biomass production of 812 mg/L of dry cells in 30 days [29]. Christwardana et al. (2020) used *Spirulina platensis* microalgae in the cathode and wastewater from a coffee shop inoculated with *Saccharomyces cerevisiae* yeast (biocatalyst) in the anode of two double-chamber MFCs with graphite electrodes to produce a voltage of 170 mV, a current density of 400 mA/m^2^, a maximum power density of 98 mW/m^2^, and a biomass of 740 mg/L [30]. Finally, Cui et al. (2014) worked with *Chlorella vulgaris* biomass in the cathode and *Scenedesmus* dry biomass with activated sludge in the anode of four double-chamber MFCs with cloth and carbon fiber electrodes, achieving a maximum voltage of 620 ± 13 mV, a power density of 1926 ± 21.4 mW/m^2^, and a biomass of 1247 ± 52 mg/L [31].

The objective of this research was to evaluate the potential of *Chlorella* sp. biomass in electricity production using single-chamber MFCs at the laboratory scale. To this end, monitoring of voltage (mV), electric current (mA), current density (mA/cm^2^), power density (mW/cm^2^), and internal resistance (R_int_, Ω) of MFCs was carried out; pH, electrical conductivity (EC, µS/cm), oxidation–reduction potential (ORP, mV), cell density (cells/mL), and optical density (OD) of the biomass were also tracked over 20 days of operation; scanning electron microscopy (SEM) and energy-dispersive X-ray spectroscopy (EDS) analyses were conducted to examine the morphology and composition of the bacterial biofilm adhered to the anodic electrodes at the end of the experiments. Despite the growing interest in MFCs, there are few studies exploring the specific use of *Chlorella* sp. Previous research has not optimized the cultivation conditions to maximize biomass production and energy efficiency in MFCs. This study provides valuable data on the energy performance and viability of *Chlorella* sp. in MFCs, promoting the development of clean energy technologies. 

## 2. Materials and Methods

### 2.1. Preparation of Culture Medium for Chlorella *sp.*

The culture medium for the microalgae *Chlorella* sp. was prepared following the procedure described by De la Cruz-Noriega et al. (2021). A culture medium based on urea (CH_4_N_2_O, 0.15 g) and potassium chloride (KCl, 0.015 g) was prepared in 1000 mL of fish waste broth previously filtered and autoclaved at 121 °C for 15 min [32]. A volume of 750 mL of *Chlorella* sp. biomass was provided by the Research Institute in Science and Technology of Cesar Vallejo University, Trujillo, Peru. The strain number or specific strain designation for *Chlorella* sp. was not specified, but biomass was used in previous research [32,33].

### 2.2. Construction of MFC 

Three single-chamber MFCs with air cathodes were fabricated. This configuration was chosen due to the positive reports on energy production with *Chlorella* sp. biomass in previous studies, in addition to the low manufacturing costs, as it does not require a proton exchange membrane and is easily reproducible for future research [33]. Hermetic polypropylene containers with a capacity of 680 mL were used. The electrodes utilized were activated carbon (AC) (anode) and zinc (Zn) (cathode) with a surface area of 20.50 cm^2^. The AC electrodes were fabricated following the procedure outlined by Agüero-Quiñones et al. (2023) and were selected for their excellent biocompatibility with microorganisms, as well as their good conductivity for electricity generation [34]. The electrodes were placed inside the cells connected with copper (Cu) wires to a closed circuit with a 100 Ω resistor. The cathodic electrode was adhered to the lateral wall of the cell, where a circular hole with a diameter of 4.5 cm was made using an adhesive sealant (Siloc 400484, Buenos Aires, Argentina) to prevent substrate leakage, while the anodic electrode was situated at the other end. Two circular holes with a diameter of 1 cm were made in the MFC lid near the edges for the copper wire exit, and a 1.5 cm diameter hole was made in the central part for the supply of O_2_ and CO_2_ through a hose connected to an aeration pump (see Figure 1). 

### 2.3. Inoculation of Chlorella sp. and Experimental Configuration of the MFC

Each MFC was filled with 200 mL of the prepared culture medium and 200 mL of pure *Chlorella* sp. biomass with an initial cellular density of 8.8833 × 10^6^ ± 5.35 × 10^5^ cells/mL and a pH of 7.10 ± 0.03. Additionally, 100 mL of synthetic wastewater was injected, resulting in a total working volume of 500 mL. Oxygen (O_2_) and carbon dioxide (CO_2_) were supplied to the microalgae using a mechanical aeration pump (Sobo SB-248A—5 L/min, Zhongshan, China), which also helped to keep the microalgae in suspension. Artificial illumination was provided through an LED light bulb (Philips 9W E27—White Light, Amsterdam, The Netherlands) operating on a 12/12 photoperiod for 20 days. The light intensity was 950 lumens. All cells were maintained at ambient temperature (21 ± 1.5 °C).

### 2.4. Monitoring of Physicochemical Parameters of Chlorella sp. Biomass

The monitoring of key physicochemical parameters of the *Chlorella* sp. biomass incorporated into the MFC was conducted. This involved measuring pH and electrical conductivity (EC) with the EZ-9909 Digital Meter and oxidation–reduction potential (ORP) with the PT-380 Portable Meter. Measurements were taken daily at ambient temperature (22 ± 1.5 °C) over the course of 20 days. 

### 2.5. Monitoring of Electrical Production of the MFC 

Daily monitoring of voltage (mV) and electric current (mA) was carried out using a calibrated multimeter (Truper MUT-830, Campeche, Mexico). The internal resistance (R_int_) was calculated using polarization curves, plotting electric current against voltage and applying a linear fit (y = mx + b), where the slope (m) represents the R_int_ of the MFC [35]. The current density (CD) and power density (PD) were obtained following the methodology of Rojas-Flores et al. (2022), using external resistances of 1, 10, 50, 100, 200, 300, 500, 750, 800, and 1000 Ω and calculated using the following formulas [36]:(1)$$\mathrm{Power}\;\mathrm{density}\;\left({\mathrm{mW}/\mathrm{cm}}^{2}\right)=\frac{\mathrm{Current}x\;\mathrm{Voltage}}{\mathrm{Surface}\;\mathrm{area}\;\mathrm{of}\;\mathrm{the}\;\mathrm{electrode}\;}$$
(2)$$\mathrm{Current}\;\mathrm{density}\;\left({\mathrm{mA}/\mathrm{cm}}^{2}\right)=\frac{\mathrm{Current}}{\mathrm{Surface}\;\mathrm{area}\;\mathrm{of}\;\mathrm{the}\;\mathrm{electrode}\;}$$

### 2.6. Measurement of Optical Density of Chlorella sp. Biomass

Optical density (OD) was evaluated by measuring absorbance using a spectrophotometer (Jenway UV/VIS-6305, London, UK) at the beginning and end of the treatment. It was measured in this way to avoid manipulating and/or contaminating the medium and to have a reference of the increase in the microalgae cell density. A sample of 3.5 mL of biomass was added to glass cuvettes with a 1:2 dilution for a total of three samples. The sample was mixed using a Vortex Shaker (Isolab-622.01.001, Eschau, Germany) at 300 rpm for 10 min to ensure homogeneity before measurement. The measurement was then performed by calibrating the equipment’s wavelength to 600 nm. 

### 2.7. Determination of Cell Density of Chlorella sp.

Cell density was determined by manual cell counting using a Neubauer chamber (0.10 mm depth, 0.0025 mm^2^ quadrant area, Kyntel, Lima, Peru) at the beginning and end of the treatment, similar to the absorbance measurements. For this, 3 samples of 15 mL of *Chlorella* sp. biomass were taken from each MFC into centrifuge tubes, and a 1:20 dilution with distilled water was performed. The samples were mixed at 3000 rpm for 10 min using a Vortex Shaker (Isolab-622.01.001, Eschau, Germany). A volume of 10 µL of the cell suspension was loaded into the chamber, and counting was performed in duplicate under a 40× objective LED Binocular Microscope (Olympus Model CX23LFS1, Tokyo, Japan). This procedure was repeated three times to ensure data accuracy, and at least 400 cells were counted per chamber to achieve an error of ±10%. The cell count values obtained were substituted into the following equation to determine the microalgae cell concentration [37]: (3)$$\mathrm{Cell}\;\mathrm{density}\;\left(\mathrm{cells}/\mathrm{mL}\right)=\frac{\mathrm{Total}\mathrm{number}\;\mathrm{of}\;\mathrm{cells}\;}{\mathrm{Number}\;\mathrm{of}\;\mathrm{square}\;}\times 10,000\times \;\mathrm{Dilution}$$

### 2.8. Physicochemical Characterization Analysis of Anodic Electrodes in MFC

Scanning electron microscopy (SEM) analysis was performed to visualize the structure of the microalgae biofilm adhered to the anodic electrodes, as well as to understand its morphology and interaction with the electrode surface. Additionally, energy dispersive X-ray spectroscopy (EDS) analysis was conducted to determine the elemental composition of the electrode that interacted with the formed biofilm. 

## 3. Results and Discussion

### 3.1. Electricity Production with Chlorella sp. in MFCs

Figure 2a shows the voltage values produced by the biomass of *Chlorella* sp. in MFCs reporting an average of 1098.80 ± 126.35 mV over the 20 days of operation. The voltage increased from the first day (1153 ± 3.61 mV) to day 4, where the highest voltage peak (1271 ± 2.52 mV) was obtained. In the subsequent days, the voltage production experienced a progressive decrease until day 20, when 899 ± 2.00 mV, the lowest recorded value, was generated. The values obtained during the first three days are assumed to be due to the microalgae potentially being in the lag phase, as they adapted to the new environmental conditions. This assumption is based on similar behavior observed in electrogenic bacteria in wastewater, which proliferate in the medium [38]. However, it is important to note that this phase was not directly monitored in this study. It has been reported that the adaptation of the microalga *Chlorella* sp. to a suitable environment involves the proper absorption of light and supplied CO_2_ to carry out photosynthesis efficiently, resulting in the production of oxygen and organic compounds, such as sugars and amino acids [39]. Some of these compounds may be released into the medium either through excretion or as a result of cell lysis. These compounds are degraded by the electrogenic bacteria (*Shewanella* sp., *Geobacter* sp., *Rhodoferax ferrireducens*, *Rhodopseudomonas palustris*), increasing electron production and, consequently, voltage [40]. As a result of this process, a progressive increase in voltage data was observed over the days. On days 4, 5, and 6, relatively high data were observed. This increase in data suggests that the microalgae were experiencing high metabolic activity [41]. It has been reported that during this stage, microalgae grow steadily and stimulate the production of significant amounts of organic matter and oxygen, providing optimal conditions for the bacteria [42]. At this point, the balance between organic matter production, electron release, and cathodic efficiency might be at its optimal level, resulting in the highest recorded voltage (1271 ± 2.52 mV). As the microalgae progressed toward the later stages of the experiment, their photosynthetic activity decreased due to suboptimal environmental conditions, such as nutrient depletion and accumulation of toxic byproducts [43]. This led to a gradual reduction in voltage until the end of the experiment on day 20. The generated voltage data exceed those obtained by Qin et al. (2024), who used a double-chamber MFC with photosynthetic microalgae biomass (*Chlorella vulgaris*, *Chlorella* sp., *Tetradesmus obliquus*, and *Microcystis aeruginosa*) as a substrate in the cathode and synthetic wastewater in the anode, achieving a maximum average voltage of 459.5 mV with the species *Chlorella vulgaris*. They indicated that voltage production was strongly linked to the dissolved oxygen (DO) of the substrate, and thus, increasing DO concentration resulted in an increase in output voltage [44]. Similarly, Qin et al. (2022) used a double-chamber MFC assisted with *Chlorella vulgaris* in the cathodic chamber and anaerobic activated sludge in the anodic chamber, producing a maximum voltage of 780 mV. They suggested that voltage production fluctuations were related to the growth cycle of the microalgae and that the oxygen generated during photosynthesis acted as an electron acceptor, allowing greater catalytic kinetic activity for voltage generation [45]. Yang et al. (2021) obtained a maximum voltage of 675 mV using the biomass of *Chlorella pyrenoidosa* in a single-chamber MFC, indicating that the obtained voltage suggested superior polarization performance with other evaluated strains due to the cell’s low internal resistance [46]. 

It is important to mention that the dissolved oxygen (DO) in the substrate plays a crucial role in the electrochemical and biological processes that enable electricity generation in MFCs [47]. Although DO was not directly measured in this experiment, the oxidation–reduction potential (ORP) values can provide insight into the oxygen dynamics within the system. High ORP values, observed during the initial days of the experiment, suggest a more oxidizing environment, which can be attributed to the photosynthetic activity of *Chlorella* sp., producing oxygen as a byproduct and increasing the amount of dissolved oxygen in the medium. This oxygen is essential for maintaining a suitable environment for aerobic microorganisms to be present [48]. Research indicates that that the DO acts as a final electron acceptor; when microorganisms consume organic matter and release electrons at the anode, the DO at the cathode acts as the final acceptor in the reduction reaction, producing water as the end product [49]. Therefore, a higher concentration of DO, inferred from high ORP values, can improve the energy efficiency of the MFC by providing a more effective electron acceptor at the cathode, resulting in greater electricity production [50], as observed in the obtained results. However, very high levels of DO can inhibit the activity of certain anaerobic microorganisms, potentially reducing the overall efficiency of the system [47]. Similarly, low levels of DO can limit the reaction rate at the cathode, also affecting electricity generation [51,52].

In Figure 2b, the values of electric current produced by the *Chlorella* sp. biomass in the MFC are observed. There was an increase in current from the first day, with an initial value of 4.55 ± 0.03 mA, up to the third day, where the peak current of 4.77 ± 0.02 mA was reached. From there, the produced current started to decrease steadily, with small increments on days 8 (4.31 mA ± 0.03) and 14 (3.36 mA ± 0.04), until reaching 3.18 ± 0.02 mA on day 15, which was the minimum value reached. An average of 3.84 ± 0.55 mA was observed over the 20 days evaluated. In the subsequent days, there was a fluctuation in current generation, with an increase on day 18 to 3.33 ± 0.04 mA and finally a reduction to 3.18 mA ± 0.04 on day 20. During the initial phase of electric current increase (days 1–3), it is presumed that the initial metabolic activation of *Chlorella* sp. and the electroactive bacteria present, along with the abundant availability of organic substrates from the culture medium and dissolved organic matter from the microalgae, and optimal conditions for their growth and adaptability [53], as well as the effective formation of the electrogenic biofilm on the anode, contributed to the observed increase in current production until reaching the peak maximum [54]. However, during days 4–15, the progressive depletion of nutrients, accumulation of byproducts, and competition for resources could have led to a reduction in the metabolic activity of microalgae and bacteria, decreasing the production of electrons and hence generating less electric current [55]. Finally, the fluctuations observed on days 16–20 could be the result of changes in the metabolic activity of microorganisms in response to nutrient depletion, seeking to maximize energy efficiency, or there were changes in the structure and activity of the biofilm that could influence electron transfer and therefore current generation [56]. The results obtained are comparable to those shown in previous studies; for example, Rojas-Villacorta et al. (2023) achieved a maximum current value of 6.04 ± 0.21 mA using double-chamber MFCs with *Spirulina* sp. microalgae substrates (cathode chamber) and pepper residues (anode chamber), indicating that the correct adhesion of electrogenic microorganisms to the anodic electrode allowed for increased initial current values [57]. Similarly, Yadav et al. (2020) achieved a maximum current of 3.1 mA in a double-chamber MFC inoculated with *Chlorella vulgaris* as substrate in the anodic chamber and as live culture in the cathodic chamber, suggesting that mutual collaboration between bacterial and microalgal populations helps improve energy yields in the cell [58].

Figure 3a depicts the internal resistance (R_int_) of the MFC calculated using Ohm’s Law (V = IR), where V represents the voltage or potential difference produced by the cell (mV), I is the generated electric current (mA), and R is the resistance (Ω) [59]. Voltage values are plotted on the *y*-axis, while current values are on the *x*-axis. It is noted that the voltage data are directly proportional to the current values, establishing a linear fit (y = mx + b), with the slope (m) representing the internal resistance of the MFC, calculated to be 200.83 ± 0.327 Ω. The internal resistance of the MFC is influenced by the breakdown of organic compounds present in the substrate, carried out by microalgae and electroactive bacteria [60]. This oxidation process occurs at the anode and releases electrons that can flow unrestrictedly through the system when the internal resistance is low [61]. However, during this process, organic residues and metabolic byproducts can accumulate, which may inhibit microbial activity and reduce efficiency in electron transfer, resulting in an increase in internal resistance [62], as observed in this research. Rojas-Villacorta et al. (2023) reported an R_int_ value of 83.784 ± 7.147 Ω using *Spirulina* sp. in a double-chamber MFC, mentioning that the low value obtained is attributed to the metallic electrodes used, which exhibit low resistance to current flow due to their characteristics [57].

On the other hand, the values of power density (PD) according to the current density (CD) are shown in Figure 3b, presenting a maximum PD of 274.514 mW/cm^2^ at a CD of 0.551 mA/cm^2^. The PD and CD are direct measures of the amount of electricity being generated in the MFC. An increase in these values suggests higher efficiency in converting the chemical energy of organic substrates into electricity [63]. Several factors can influence the values of PD and CD in MFCs, such as the quality of the organic substrate, photosynthetic activity, microbial diversity, electrode composition, temperature, and pH of the medium, among others [64,65]. In other research, such as the study conducted by Sallam et al. (2021), a maximum power density (PD) of 0.25 mW/cm^2^ was achieved using a dual-chamber MFC inoculated with *Spirulina platensis* in the anode chamber and seawater in both the anode and cathode. They asserted that the photosynthesis-driven oxygenation by algae in the anode increased electron availability on the cathode surface, leading to enhanced energy performance in the cell. [66]. Additionally, Wang et al. (2018) obtained a maximum PD of 0.000406 mW/cm^2^ and a CD of 0.004634 mA/cm^2^ using the microalgae *Chlorella* in single-chamber MFCs, indicating that the photosynthetic biocathode improved power density values in the system [67]. 

### 3.2. Monitoring of pH, Electrical Conductivity (EC), and Oxidation–Reduction Potential (ORP) of Chlorella sp. Biomass

Figure 4a displays the monitored pH values of the *Chlorella* sp. biomass. A sharp increase was observed from the first day with a value of 7.32 ± 0.03, reaching 8.03 ± 0.02 on the eighth day. Subsequently, the increase was less pronounced with small fluctuations until day 16, where a value of 8.05 ± 0.04 was recorded. In the following days, a decrease was noticed, reaching a value of 7.74 ± 0.02 on day 20. pH is a fundamental parameter that affects both the efficiency of the MFC and the health and metabolic activity of microalgae and other microorganisms involved in the process [30]. The sharp increase from the first day to the eighth day (7.32 ± 0.03 to 8.03 ± 0.02) can be mainly attributed to the photosynthetic activity of microalgae, as they used artificial light to convert CO_2_ and water into glucose (C_6_H_12_O_6_) and oxygen [68]. During this process, CO_2_ from the medium was absorbed by the microalgae, reducing its concentration and hence decreasing the formation of carbonic acid (H_2_CO_3_) in the substrate, which contributed to the pH increase [69]. While the microalgae continued to absorb CO_2_, this process did not necessarily indicate that they were at the peak of their metabolic activity throughout the entire period, but rather that they maintained photosynthetic activity to some extent. On the other hand, during the less noticeable increase with small fluctuations until day 16 (8.05 ± 0.04), it is possible that the photosynthetic activity of *Chlorella* sp. remained, but at a more constant rate. Additionally, the consumption of nutrients from the culture medium, such as potassium chloride, could release alkaline ions such as potassium (K⁺), which increased the pH. The observed decrease until day 20 (7.74 ± 0.02) may be due to the depletion of nutrients available in the medium, which could have affected the ability of the microalgae to maintain a high rate of photosynthesis [70]. Alternatively, this decrease could also be attributed to the respiratory activity of the microalgae and bacteria, which increases the formation of carbonic acid and consequently raises the concentration of hydrogen ions (H⁺). Additionally, it could also be attributed to a high rate of proton production compared to the exchange rate towards the cathode, resulting in proton accumulation at the anode [71]. In other studies, variations in pH values have been observed due to the particular composition of each substrate. For example, Ribeiro et al. (2022) found increases in pH (~7 to 9) in the biomass of *Escherichia coli* with *Desmodesmus subspicatus*, *Pseudomonas aeruginosa* with *Desmodesmus subspicatus*, and *Escherichia coli* with *Pseudokirchneriella subcapitata* inoculated into three MFCs after seven days due to the formation of OH⁻ ions by the slowed migration of H⁺ ions from the anode to the cathode. Nonetheless, the pH range remained between values of 6 and 8, which was ideal for microbial growth [72], similar to those presented in this study. 

In Figure 4b, the recorded values of the electrical conductivity of the *Chlorella* sp. biomass incorporated into the MFC can be observed. A noticeable decrease was observed from the first day, where it peaked at 2450 ± 17.1 µS/cm, to day 11, where a value of 1836 ± 29 µS/cm was recorded. In the following days, the values showed fluctuations until reaching a value of 1969.5 ± 17 µS/cm on day 20. High values of electrical conductivity indicate a greater capacity of the medium to conduct electricity, which is beneficial for electron transfer efficiency and electricity generation in the MFC [73]. The high peak of conductivity recorded on the first day can be explained because at the beginning of the experiment, the culture medium contained high concentrations of nutrients, which had a variety of dissolved ions, such as K^+^, NH_4_^+^, and NO_3_^−^, distributed homogeneously in the solution, providing high initial conductivity [74]. On the other hand, the progressive decrease observed until day 11 is because during this time, *Chlorella* sp. and electroactive bacteria were in an active growth phase, rapidly consuming the nutrients and ions present in the medium; therefore, the number of ions decreased, progressively reducing the EC [75] and reflecting a decrease in electricity generation. Likewise, the formation of the biofilm on the anodic electrode could influence the quantity of ions in the solution because these were used in the metabolic processes of the adhered cells, reducing their availability in the medium [76] and, consequently, the conductivity. Finally, the fluctuations observed until day 20 are attributed to the variability in the release and consumption of ionic metabolites by the microalgae and bacteria, as metabolic activity may not have been constant, with periods of greater and lesser ion release as a result of changes in photosynthetic activity [77]. 

As for the oxidation–reduction potential values of the biomass, as shown in Figure 4c, these exhibited an increase from the first day, with a value of 633 ± 26 mV, until reaching their peak of 952 ± 20 mV on the third day. Subsequently, these values began to gradually decrease, experiencing fluctuations, until reaching a value of 249 ± 23 mV on the 20th day. High ORP values indicate a more oxidizing environment, allowing certain microorganisms to thrive under these conditions, increasing electron production at the anode and thereby favoring electricity generation in the MFC [78]. During the initial days, the increase in ORP, from 633 ± 26 mV to 952 ± 20 mV, can be attributed to the intense photosynthetic activity of *Chlorella* sp., producing oxygen as a byproduct and increasing the amount of dissolved oxygen in the medium [79]. This oxygen acts as an oxidizing agent in the system, which is favorable for the activity of aerobic and electroactive microorganisms, initially contributing to a higher electron production at the anode and, therefore, a higher redox potential [80]. As photosynthesis continues, microalgae and bacteria consume nutrients, reducing their availability. Additionally, as the microalgae grow and release oxygen and dissolved organic matter, bacteria also proliferate and consume the oxygen. This indicates that while photosynthesis may not have ceased, increased respiration by microorganisms could have occurred, leading to microbial competition and a decrease in overall metabolic activity [81]. This gradual reduction in metabolic activity results in a decrease in ORP. The variation observed from the 3rd day until the 20th day is due to the depletion of nutrients and the accumulation of metabolic products. As nutrients become depleted and metabolic byproducts accumulate, the environment becomes less oxidizing and more reducing, leading to a decrease in ORP [82].

### 3.3. Evaluation of Optical Density and Cell Density of Chlorella sp. Biomass

Figure 5a presents the initial and final values of optical density, measured by absorbance at 600 nm. A noticeable increase is observed after 20 days of operation, with the average value rising from 2.918 ± 0.004 AU in the three MFCs on the first day to 3.471 ± 0.195 AU at the end of the experiment. The increase in optical density can be explained by the outstanding growth and proliferation of microalgae in a suitable environment [83], under optimal conditions of light, nutrients, and temperature, favoring efficient photosynthesis [84,85]. Additionally, the high initial availability of essential nutrients, such as nitrogen from urea and other micronutrients from the fish waste broth, promotes cell growth, allowing rapid multiplication of *Chlorella* sp. cells and, consequently, an increase in biomass [86], reflected in an increase in optical density as observed in this research. Therefore, optical density is used as an indirect but quantitative indicator of microalgae growth in the system. By monitoring optical density, microalgae growth can be evaluated, and, consequently, the energy efficiency of the MFC can be inferred [87]. 

In Figure 5b, the cell density of *Chlorella* sp. biomass at the beginning and end of the experiment in the three MFCs is observed. A remarkable growth in the number of cells is evident, increasing from an average of 8.8833 × 10^6^ ± 5.35 × 10^5^ cells/mL on the first day in the three cells to 2.2933 × 10^7^ ± 1.15 × 10^6^ cells/mL on day 20. The increase in cell density can be explained by the availability of nutrients in the culture medium, providing microbial cells with the necessary elements for growth and reproduction [88]. Additionally, the appropriate pH of the medium (7.32 ± 0.03 to 7.74 ± 0.05) and optimal light conditions within the MFCs favored the metabolic activity of the microalgae, promoting healthy cell growth [89]. Furthermore, the architecture of the MFCs provided a conducive environment for the growth of microalgae by offering an interface between the liquid phase of the culture medium and the air, facilitating the absorption of CO_2_ and the release of oxygen during photosynthesis, thus stimulating cell proliferation [90]. Lastly, the presence of conductive and biocompatible carbon-based electrodes provided a suitable surface for the adhesion and proliferation of microalgae, improving the efficiency of electron and nutrient exchange [15]. Cell density can influence electricity generation in MFCs. An increase in cell density can increase the amount of organic matter metabolized, which in turn could stimulate the production of electrons available for transfer through photosynthesis [91]. However, an excess of cell density could lead to competition for nutrients and space, limiting metabolic efficiency and ultimately reducing electricity production in the cell. Ullah (2024), in their research on double-chamber photosynthetic MFCs, observed an increase in the concentration of *Scenedesmus* sp. from 1500 mg/L to 5165 mg/L within 96 h, mentioning that the remarkable growth had a significant effect on energy generation in the cell, which was driven by a high nutrient rate present in the medium [92]. 

### 3.4. Physicochemical Characterization of the Anodic Electrodes of MFCs Using SEM and EDS

In Figure 6, SEM images of the activated carbon anodic electrodes after the operating period in the MFC are presented. Figure 6a shows the electrode surface with a scale bar of 1 mm, highlighting the presence of pores distributed throughout the surface, consistent with the typical porous structure of this material. The pores provide a larger surface area and facilitate the adhesion and growth of bacterial and microalgal biofilms, promoting the formation of a favorable interface for electron transfer in the MFC [93]. Additionally, a smooth texture of the electrode surface is observed, suggesting that there was no excessive accumulation of materials or obstructions that could inhibit biofilm adhesion. On the other hand, Figure 6b provides a detailed view of the dispersed pores and solid activated carbon particles across the surface (1 mm), indicating multiple anchoring sites for biofilm adhesion, which promoted dense and uniform colonization of microorganisms [94]. The micrograph also highlights the presence of biofilm residues, consistent with the presence of attached microorganisms and the deposition of metabolic products, suggesting active metabolic activity in the MFC during the operation period. On the other hand, in Figure 6c, remnants of the *Chlorella* sp. biofilm adhered to the anodic electrode (500 μm) are evident, indicating an active interaction between the microalgae and the electrode surface [95], where the filamentous morphology suggests a complex structural organization and a firm adhesion to the electrode surface. Finally, in Figure 6d, complete adhesion of the *Chlorella* sp. biofilm to the CA electrode after the 20-day experiments in the MFC (50 mm) is observed. This indicates successful colonization of the microalgae on the electrode surface. Furthermore, the accumulation of biomass deposited at the bottom of the cell suggests high metabolic activity and active proliferation of microalgae during the operation of the cells. A well-developed biofilm provides a larger reaction surface and higher metabolic activity, improving the efficiency of electricity generation in the MFC [96].

Figure 7 shows the EDS spectra analysis of the activated carbon anode electrode that interacted with the *Chlorella* sp. biofilm, reporting the composition of the present elements. The predominant elements are oxygen (O), carbon (C), silicon (Si), aluminum (Al), iron (Fe), sodium (Na), magnesium (Mg), potassium (K), calcium (Ca), and chromium (Cr). The high presence of oxygen and carbon is expected, as they originate from both the activated carbon electrode material, the redox processes in the cell, and the *Chlorella* sp. biomass [17,97]. The presence of aluminum is due to the mesh of this material within the electrode [34]. The presence of sodium, magnesium, potassium, and calcium can be attributed to the dissolved salts in the cultivation medium, as they are present in the form of dissolved ions in mineral salts [74]. Finally, the detection of chromium, silicon, and iron can be attributed to the composition of the synthetic wastewater used [98]. Identifying the elemental composition helps to better understand the interaction between the microalgae and the electrode, which is essential for optimizing operational conditions and improving the system’s efficiency. 

## 4. Conclusions 

This research successfully demonstrated that the biomass of *Chlorella* sp. has a high potential for sustainable electricity production in single-chamber MFCs, achieving outstanding voltage values of 1271 ± 2.52 mV, surpassing those previously reported by other researchers. The biomass of *Chlorella* sp. increased to 2.2933 × 10^7^ ± 1.15 × 10^6^ cells/mL after 20 days of operation, maintaining optimal pH values for its growth between 7.32 ± 0.03 and 7.74 ± 0.05, with peaks of EC at 2450 ± 17.1 µS/cm, indicating a high capacity of the medium for electron transfer, and high ORP values of 952 ± 61 mV, demonstrating an oxidizing environment in the system. SEM analysis allowed for the observation of filamentous structures of the *Chlorella* sp. biofilm adhering to the activated carbon electrode during the process. Additionally, EDS spectra reported the predominant presence of oxygen and carbon. This study explored the impressive potential of *Chlorella* sp. biomass to be used on a large scale as fuel for bioelectricity generation in MFCs, contributing to the development of sustainable technologies for green energy generation and the diversification of renewable energy sources to achieve the energy transition.

For future research, it is recommended to include the measurement of dissolved oxygen in the culture medium to assess its influence on the growth and performance of *Chlorella* sp. in MFCs. This variable is crucial for understanding optimal cultivation conditions and enhancing process efficiency. Additionally, it is essential to conduct detailed studies of the growth curve of *Chlorella* sp., as understanding the different growth phases will allow for the optimization of biomass harvest timing to maximize energy yield. Likewise, optimizing cultivation conditions such as light intensity, substrate type, and nutrient levels is recommended to improve biomass production and metabolic efficiency. Finally, exploring the energy potential of different *Chlorella* sp. strains is advisable to identify those with the best performance in MFCs. Comparing strains could reveal specific characteristics that optimize electricity production and cell stability.

## Figures and Tables

**Figure 1 bioengineering-12-00635-f001:**
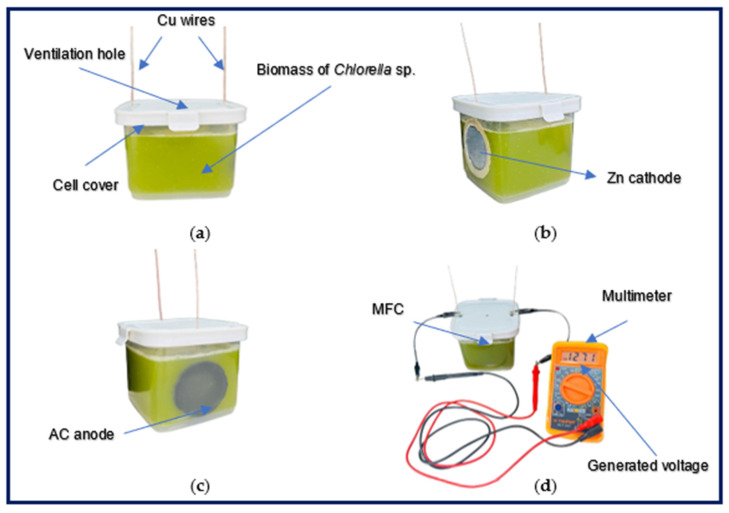
Microbial fuel cells (**a**) prototype, (**b**) cathodic electrode, (**c**) anodic electrode, and (**d**) electricity generation.

**Figure 2 bioengineering-12-00635-f002:**
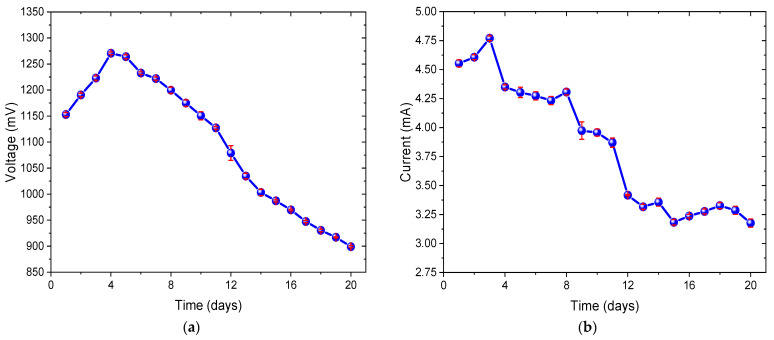
Monitoring of (**a**) voltage and (**b**) electric current values of the MFC.

**Figure 3 bioengineering-12-00635-f003:**
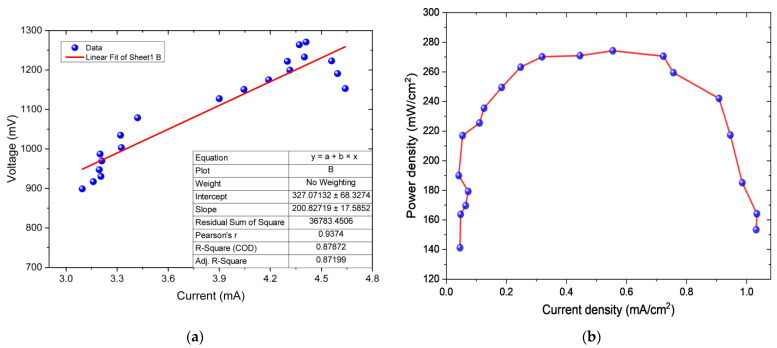
Values of (**a**) internal resistance and (**b**) power density as a function of current density in the MFC.

**Figure 4 bioengineering-12-00635-f004:**
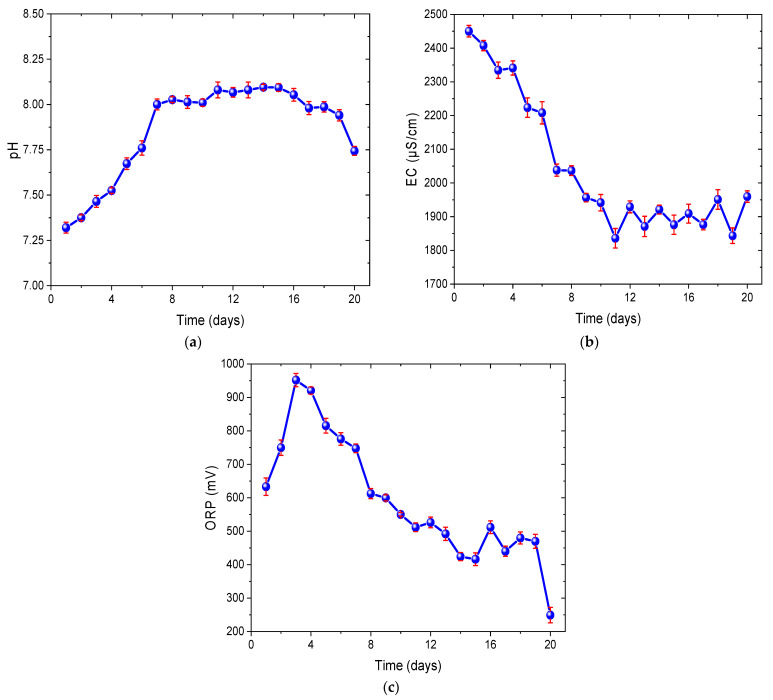
Monitoring of (**a**) pH, (**b**) EC, and (**c**) ORP of the biomass supplied to the MFC.

**Figure 5 bioengineering-12-00635-f005:**
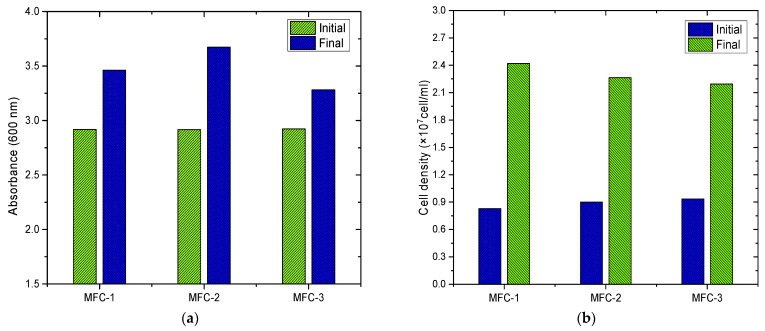
Initial and final variation of (**a**) absorbance and (**b**) cell density of *Chlorella* sp.

**Figure 6 bioengineering-12-00635-f006:**
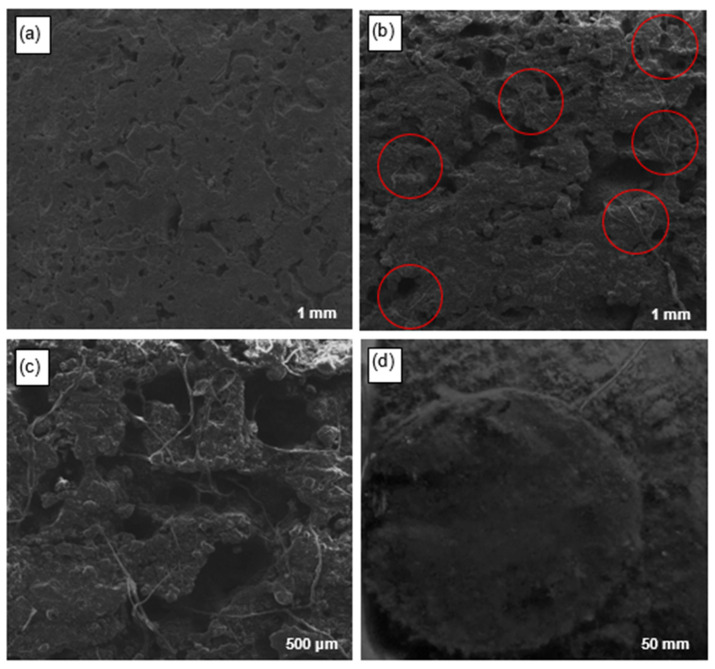
Micrographs of the anodic electrodes: (**a**) surface, (**b**) CA pores, (**c**) morphology of the attached biofilm, and (**d**) biofilm attached at the end of treatment in the MFC.

**Figure 7 bioengineering-12-00635-f007:**
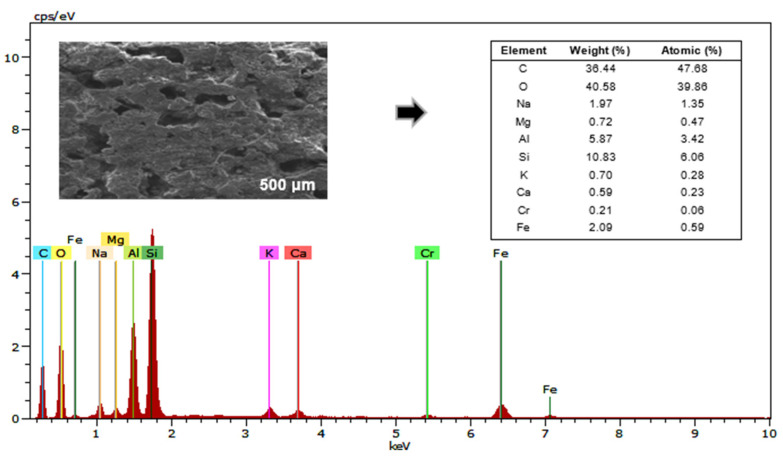
EDS spectra of the anode electrodes at the end of the treatment in the MFC.

## Data Availability

The original contributions presented in the study are included in the article; further inquiries may be directed to the corresponding author.

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
