# Peer review of "Electric Potential of Chlorella sp. Microalgae Biomass in Microbial Fuel Cells (MFCs)"

_bioengineering, 2025, doi:10.3390/bioengineering12060635_

Round 1
Reviewer 1 Report (Previous Reviewer 1)
Comments and Suggestions for Authors
Dear authors,
Thank you for this revised version of your manuscript. Most of my previous comments were promptly addressed, so I only have minor comments.
Line 4: What happened to Segundo Rojas-Flores?
Line 117: When I mentioned that De la Cruz-Noriega et al. (2021) was in Spanish, I meant the article... I know that the author's full surname can not be translated. You state that the culture medium was prepared accordingly to the cited reference. I asked you to develop this section as non Spanish readers will not understand a procedure written in Spanish.
Figures 2 and 4: State that values presented are mean ± SD.
Line 320: Reference 67 is missing.
Author Response
REVIEWER 1
Dear authors,
Thank you for this revised version of your manuscript. Most of my previous comments were promptly addressed, so I only have minor comments.
Line 4: What happened to Segundo Rojas-Flores?
Answer: Dr. Segundo Rojas Flores has decided to withdraw as co-author of the manuscript due to work commitments and restrictions from another university where he works to avoid conflicts of affiliation, due to which the author voluntarily decided not to participate in the publication. The same case occurred with Dr. Jose Cruz Monzon. We appreciate his contribution during the stages of the project. All authors have kindly accepted the changes made.
Line 117: When I mentioned that De la Cruz-Noriega et al. (2021) was in Spanish, I meant the article... I know that the author's full surname can not be translated. You state that the culture medium was prepared accordingly to the cited reference. I asked you to develop this section as non Spanish readers will not understand a procedure written in Spanish.
Answer: The specific procedure for preparing the culture medium from De La Cruz-Noriega et al. (2021) was added for better understanding for English readers.
Figures 2 and 4: State that values presented are mean ± SD.
Answer: Indeed, the figures in question show the mean of the monitored values ​​with their respective standard deviation, as previously requested by the reviewers.
Line 320: Reference 67 is missing.
Answer: There was a typing error. Reference 67 has been added.
Reviewer 2 Report (Previous Reviewer 2)
Comments and Suggestions for Authors
Some suggestions
For last paragraph of Introduction section, please explain which gap in the literature this study fills.
For sections 2.6 and 2.7, did you use any statistical method? If not, why?
In experiment, why did you not measure the dissolved oxygen concentration?
I suggest that authors should systematize and point out the most important conclusions in summary.
It could be enhanced by providing specific recommendations for future research.
Author Response
REVIEWER 2
Some suggestions
For last paragraph of Introduction section, please explain which gap in the literature this study fills.
Answer: Added explanation of the literature gap that this research will fill, as requested.
For sections 2.6 and 2.7, did you use any statistical method? If not, why?
Answer: No statistical methods were used, only average values ​​for the corresponding parameters were used. No specific statistical analysis was used because we were only interested in obtaining the average values ​​to later analyze their behavior, but we did not have to relate or predict results. However, statistical methods can be incorporated if we were to compare results with other systems to see statistical differences or similarities.
In experiment, why did you not measure the dissolved oxygen concentration?
Answer: The dissolved oxygen concentration was not measured due to lack of necessary equipment. However, we acknowledge the importance of measuring DO and plan to include this measurement in future studies to provide a more comprehensive analysis of the culture conditions.
I suggest that authors should systematize and point out the most important conclusions in summary.
Answer: The writing of the conclusions was systematized and restructured to point out the most important findings of the research.
It could be enhanced by providing specific recommendations for future research.
Answer: Specific recommendations for future research that arose from this research were added.
This manuscript is a resubmission of an earlier submission. The following is a list of the peer review reports and author responses from that submission.
Round 1
Reviewer 1 Report
Comments and Suggestions for Authors
Abstract: Please, state the meaning of the abbreviations employed (CA and UA). No need for presenting 4 figures on line 23.
Line 78: Why referring specifically to cyanobacteria? Is it a reference for bacteria (which they are) or for microalgae (which they also are)? It is a bit confusing...
Line 99: Use the same unit to refer to biomass, you use both mg/L and g/L.
Line 100: Why is lyzed initiating with a capital L and italicized?
Line 107: You can mention the abbreviation and unit inside the same parenthesis. Ex. (EC, µS/cm).
Line 116: De la Cruz-Noriega et al. (2021) is in Spanish. Please, develop further this section as not everybody will understand Spanish.
Line 119: Is there a strain number for Chlorella sp.?
Line 122: So, in total, three repetitions were performed, right? Why you don't mention the SD in your results? No statistical analysis was performed?
Line 139: What's the initial cell density or biomass?
Line 142: Aeration was also employed to keep the microalgae in suspension, right?
Line 143: Ligth intensity was not measured?
Line 147: I would not consider only 3 parameters as various.
Line 148: Are there other parameters not mentioned here? Why ‘this included’?
Line 163: Why only ate the beginning and end of the experiment?
Line 166: Why did you centrifuged the sample to measure its absorbance? You mean that you mixed the sample, right? Why mixing it for 10 minutes?
Line 170: Which frequency? Daily? Weekly?
Line 173: Not centrifuged, but mixed.
Line 177: Was there a minimum of cells counted? To have a ±10% error, at least 400 cells must be counted per chamber.
Line 189-190: Don’t you have mean ± SD for voltage?
Line 193: How can you state that the microalgae were in lag phase if you didn’t monitored it?
Line 194: You don’t mentioned bacteria in the M&M section. You only mentioned a synthetic wastewater.
Line 198: You mean extracellular organic matter. Not necessarily the produced organic compounds get excreted into the medium.
Line 201: How can you state that if you didn’t monitored it daily? You can’t state that.
Line 203: You didn’t measured the dissolved organic matter. Those are assumptions, not statements.
Line 208: How so if you injected atmospheric air? pH stabilized at 8.2. If CO2 was exhausted, pH would be higher.
Line 235: Again, as you didn’t measured DO, you can’t make any statement about it, only assumptions. Can you relate the DO concentrations with the ORP data to support your statement?
Line 239: No SD on the figures?
Line 249: Which organic substrates do you mean? From the medium or dissolved organic matter from the microalgae?
Line 252: Do you believe that the culture achieved stationary phase after only 3 days?
Line 275: Not the same SD as presented in figure 2a.
Line 279: Which residues? Organic matter?
Line 295: Units are different (mW/cm2 vs W/m3), making it hard to compare the data.
Line 299: Convert from m to cm.
Line 312: So, microalgae experienced exponential growht phase until day 8. This undermines your statement at line 252.
Line 316: So, constant rate = stationary phase.
Line 318: Or, more simply, due to respiration.
Figure 4b: Change CE for EC.
Line 365: Or, as microalgae grows and releases oxygen and dissolved organic matter, bacteria also grows and consume the oxygen. Not necessarily that photosynthesis has stopped, but that respiration increased. Check your pH data.
Line 375: Why only two sample points?
Line 377: If you have 3 measurements, then state the mean ± SD.
Line 390: Unfortunately, as you don’t have any intermediate sample, nothing can be said about growth… the culture was growing on day 20 or was it dying? That is why I don’t agree with your previous statements on growth phases.
Line 405: Usually, cultures get light limited due to self shading before nutrient limited. Do you know the amount of nutrients in your medium?
Comments on the Quality of English Language
Line 76: Missing the 't' at this. Check the manuscript for the same issue.
Line 96: Café?
Line 189: today?
Line 230: his DO?
Line 411: Médium?
Line 417: The squares in Fig 6a and b are 1 mm by 1 mm (1 mm2 in total)?
Line 427: Please, indicate it in the figure.
Reviewer 2 Report
Comments and Suggestions for Authors
The 21% similarity in the ithenticate program should be reduced, a maximum of 15% is acceptable. On the other hand, it is an interesting topic. Some suggestions:
For section 2.2, please give the surface areas of electrodes,
For section 2.3, please write the light density for Chlorella.
Line 148, please write the long name of CE. Is it CE or EC? Line 332, Line 302 and Figure 4b, please correct them.
Line 149, please write the long name of ORP
Line 158, please add “t” before “he”
For sections 2.6 and 2.7, did you use any statistical method? If not, why?
In experiment, why did you not measure the dissolved oxygen concentration?
Line 198, please give examples for electrogenic bacteria in parenthesis.